# Comparison of the Influence of Two Types of Plasma Treatment of Short Carbon Fibers on Mechanical Properties of Epoxy Composites Filled with These Treated Fibers

**DOI:** 10.3390/ma15186290

**Published:** 2022-09-09

**Authors:** Jana Novotná, Martin Kormunda, Jakub Perner, Blanka Tomková

**Affiliations:** 1Department of Material Engineering, Faculty of Textile Engineering, Technical University of Liberec, 461 17 Liberec, Czech Republic; 2Department of Physics, Faculty of Science, University of Jan Evangelista Purkyně in Ústí nad Labem (UJEP), České Mládeže 8, 400 96 Ústí nad Labem, Czech Republic

**Keywords:** recycled carbon fiber (RCF), fiber-reinforced epoxy composites (FRE), plasma treatment, mechanical properties

## Abstract

The interfacial interface between fibers and matrix plays a key role for epoxy matrix composites and short recycled randomly arranged fibers. This study used short recycled carbon fiber (RCF) as a filler. Plasma treatment was used for carbon fiber surface treatment. This treatment was performed using radio (RF) and microwave (MW) frequencies at the same pressure and atmosphere. Appropriate chemical modification of the fiber surfaces helps to improve the wettability of the carbon fibers and, at the same time, allows the necessary covalent bonds to form between fibers and the epoxy matrix. The effect of the plasma treatment was analyzed and confirmed by X-ray photoelectron spectroscopy, Raman microscopy, scanning electron microscopy, transmission electron microscopy and wettability measurements. Composite samples filled with recycled carbon fibers with low concentrations (1 wt%, 2.5 wt% and 5 wt%) and high concentrations (20 wt% and 30 wt%) were made from selected treated fibers. The mechanical properties (impact toughness, 3PB) were analyzed on these samples. It was found that the modulus of elasticity and bending stress increase with the increasing content of recycled carbon fibers. A more significant change in impact strength occurred in samples with low concentration.

## 1. Introduction

In recent years, the production of carbon fiber-filled composites has increased significantly, and thus the topic of recycling these materials has become increasingly important [1]. When developing new materials, it is necessary to consider the life cycle of the resulting products and to pay attention to the possibilities of reusing recycled raw materials. This work focuses on the use of short recycled carbon fibers as filler in epoxy composites. The shorter the fibers used as reinforcement components in composites, the more significant the role of their cross-sectional area entering the interfacial interaction. In the composites industry, the interface between fibers and matrix is of fundamental importance for the mechanical properties of fiber-reinforced composite materials. The optimal interfacial interface between fibers and matrix depends on the specific requirements of each application, and the possibility of modifying it is significant [2]. To date, there has been no published research report addressing the effect of plasma action, particularly concerning fiber ends and their cross-sections, as in this study.

Furthermore, this study analyzes the mechanical properties of epoxy composites reinforced with recycled short carbon fibers (RCF) with and without plasma treatment. Plasma treatment usually increases the adhesion between fibers and matrix, and the incorporation of the fibers into the matrix provides an ideal reinforcement effect. The mechanical properties of the RCF-filled composites were analyzed using impact toughness and flexural strength tests. These tests were chosen because, based on previous knowledge of short fiber reinforced composites, it can be assumed that the tested specimens have low ductility. In tension, the applied tensile force would act primarily on the matrix of the material, and short fiber reinforcement would not significantly help the tensile strength.

In general, the structure of carbon fibers varies depending on their precursor, with graphitic basal planes usually present on their surface, which exhibits poor wettability and low interaction with polymers [2]. Therefore, surface treatments are applied to the surface of CFs to improve their interaction with the matrix. Plasma deposition of carbon fibers is a dry reaction process; depending on the process conditions, it can have several effects that may overlap. The reason for these treatments is to increase the roughness of the fiber surface, remove impurities from the fiber surface, remove weakly bonded carbon planes or attach chemically active elements, which allows for better interfacial adhesion in composites [2,3,4]. The action of inert gases results in crosslinking, where two or more parallel polymer chains are linked by O_2_ or He or Ar, and the by-products allow the formation of bonds to adjacent chains [5,6]. The surface properties of carbon fibers can be affected by foreign elements present on the surface of these fibers; therefore, in this study, much attention was paid to the analysis of the fiber surface using X-ray photoelectron spectroscopy (XPS).

Plasma treatment of additives can solve two basic requirements for the quality of composites: homogeneous dispersity; and a strong bond between the additive and the matrix. However, the operating conditions of plasma treatment need to be appropriately selected. For virgin CF, Morgan [7] reports the following operating conditions for a plasma reactor: 100 W, RF generator at a pressure of 2–50 Pa and an operating time of 20 s to 20 min [3,7,8]. Donnet [9] points out that different results can be achieved depending on whether the plasma is guided by radio frequency (RF) or microwave (MW).

In recycled CFs, surface changes also occur due to plasma action, as reported by [10,11]. These studies confirmed that plasma exposure causes a slight deepening of the surface roughness of recycled carbon fibers. Both studies involve a polypropylene matrix, whereas the more common epoxy matrix used in this work is an epoxy matrix. Study [9] notes a reduction in flexural strength of plasma treated RCF composite samples at higher plasma powers (200 W and 300 W) and recommends the use of 100 W for 70 s. Study [10] reports a concise plasma exposure time of 0.5 s or less at 0.8 kW at RF frequency. The experimental conditions chosen in this study are based on these data. The study by Altay [10] compares the mechanical properties of composites at 20 wt% concentration and the study by Hooseok [11] at 15% ***v*/*v*** concentration.

In a previous study [12], it was experimentally confirmed that along with the incorporation of new surface functional groups, changes in fiber topography could also occur, which will be further analyzed by scanning electron microscopy (SEM) and X-ray photoelectron spectroscopy (XPS).

## 2. Experiment

### 2.1. Materials

An epoxy matrix system made from a low viscosity bisphenol A-based epoxy resin L 285, together with a cyclic-aliphatic polyamine curing agent H 508, with a mixing ratio of 100:40 by weight, was used. The density of the resin was 1200 kg/m^3^ and the density of the hardener was 1030 kg/m^3^. Commercially available PAN-based Carbiso milled carbon fiber was used, with a bulk density of 400 g/L. The fibers were 7.0 ± 0.3 µm in diameter and 100 ± 9 µm in length. These fibers were manufactured by ELG Carbon Fiber (Coseley, UK). ELG uses dry carbon fiber waste from production and uncured prepregs and cured laminates containing CF at the end of life to produce these fibers. RCF Carbiso has been produced using a modified patented pyrolysis process. Because the fibers were recycled, they contained no sizing but may have contained trace metals (<0.5 g/1 kg).

#### 2.1.1. Plasma Treatment of Recycled Carbon Fibers

Based on the available data presented in [7,8,9,10,13,14,15], the following conditions were selected for the plasma application: source power: 30 W and 100 W, time: 0.5–50 min. This work analyses two types of plasma treatments of recycled carbon fibers. The same pressure (100 Pa) is used for both treatments, but they differ in the treatment time and the frequency used.

##### Microwave Plasma Treatment

The first type of plasma treatment was carried out at microwave frequency (MW) under the influence of oxygen 200 sccm and hydrogen 50 sccm at a power of 100 W. A particular fluidization device adapted for very light bulk materials was used to treat the recycled carbon fibers. The device contained a unique hopper that ensures uniform dosing of the bulk filler to the plasma apparatus; the plasma action lasted 1**–**50 min. Small cracks were observed in the SEM images of the recycled plasma-treated carbon fibers (exposure time ≥10 min), as shown in Figure 1c. Based on a pilot series of specimens containing these plasma-treated recycled carbon fibers (results published in [12]), where the majority of the plasma-treated fiber-filled composite specimens showed a deterioration in mechanical property values compared to the untreated fiber-filled specimens, a series of plasma-treated fibers were fabricated where the treatment time was reduced to 1 min. After this change, no more destruction was found in the SEM images of the fibers, see Figure 1b.

##### Radiofrequency Plasma Treatment

Based on the recommendation already published by Donnet [9], an apparatus for the treatment of powder fillers by radiofrequency (RF) plasma treatment was developed. The duration of this treatment was 30 s. No defects were observed in the SEM images of the recycled fibers treated by radiofrequency plasma treatment, as shown in Figure 2. The recycled carbon fibers were plasma treated in a fluidized bed reactor. The working gas (air) flowed into the reactor at its bottom, fluidizing the powder. Two outer ring electrodes were attached to the bottom of the reactor 1 cm apart to ignite the plasma discharge. The electrodes were connected to a Dressler Cesar 133 RF generator via a matching network. The airflow rate was 150 sccm.

The plasma treatment times (1xcycle = 30 s, 3xcycles = 90 s) and the applied power of 30 W and 100 W were compared. Based on the XPS analysis showing the elemental composition of the plasma-treated recycled carbon fibers shown in Figure 5 samples prepared using 100 W for 30 s were selected for application to epoxy composites.

#### 2.1.2. Characterization of Recycled Carbon Fibers

**SEM.** Characterization of the effect of plasma treatments on recycled carbon fibers was performed by scanning electron microscopy (SEM; VEGA 3 TESCAN (TESCAN ORSAY HOLDING, a.s., Brno, Czech Republic); ZEISS NEOPHOT 32 (Carl Zeiss Microscopy GmbH), Jena, Germany)

**TEM.** High-resolution transmission electron microscopy (HR-TEM) images were obtained using an FEI Titan electron microscope operating at 80 kV (Titan, Thermo Fischer Scientific, London, UK).

**Wetting measurements** were performed by determining the water contact angle (WCA). It was measured using a DSA30 droplet shape analyzer (Krüss, Berlin, Germany) with a water volume of 3 µL; RCF was spread on adhesive tape to measure the contact angle. Droplets were applied to each prepared sample and then the approximate WCA value was calculated.

**XPS analysis.** X-ray photoelectron spectra (XPS) were recorded using a Phoibos 100 hemispherical analyzer (from Specs) operated in FAT mode. An AlKα spectral line with an energy of 1486.6 eV was used and the spectra were referenced to aliphatic carbon bonds at 285 eV. Resolution spectra for a transition energy of 10 eV were used and the CasaXPS software (including RSF factor) [16] was used for calculations.

### 2.2. Preparation of FRE Composite Samples

For low-fiber concentrations (1 wt%, 2.5 wt% and 5 wt%), the epoxy, hardener and fibers were mixed for 10 min using a magnetic stirrer. At higher concentrations (20 wt% and 30 wt%), manual mixing was used. The plasma-treated RCF could be more easily mixed into the epoxy-hardener mixture, and magnetic stirring was also used for the 20 wt% concentration. The mixture was then poured into molds and left at room temperature for 24 h. Subsequently, the mixture was left in the oven for 15 h at 60 °C. The overall sample preparation process is presented in [17].

### 2.3. Characterization of FRE Composite Samples

**SEM analysis.** Characterization of the fracture surfaces of the FRE composites was performed by scanning electron microscopy (SEM; VEGA 3 TESCAN); the images are shown in Figure 6.

**The impact toughness** was tested according to ASTM D 256. A schematic of the test equipment is shown in Figure 3. A 2.7 J pendulum was used at an impact velocity of 3.46 m^2^/s. The dimension of each specimen was 100 × 10 × 3 mm. The specimen thus prepared was placed on two supports with the larger specimen facedown and then broken with a pendulum hammer on the narrower side of the “edgewise” body. Samples with the same fiber concentration were measured 10 times, and the arithmetic mean and standard deviation was calculated from the data obtained. The impact toughness measurements A [J/m^2^] are shown in Figure 8. The following formula was used for the calculation:(1)$$A=\frac{E_{c}}{h\cdot b\;},$$
where $E_{c}$ [J] is the impact energy required to break the sample, *h* [m] is thickness and *b* [m] is width of the composite samples.

**The flexural strength** was measured according to EN ISO 14125. A schematic of the test equipment is shown in Figure 4. The three-point bending test was carried out on a Tiratest 2300 testing machine. The dimension of each specimen was 100 × 10 × 3 mm. The movement of the standing transom, which applied the deformation to the test specimen, was set to a maximum displacement of 1.5 mm. Samples with the same fiber concentration were measured 10 times, and the arithmetic mean and standard deviation were calculated from the data obtained. The resulting bending stress 𝜎 [MPa] and elastic modulus 𝐸 [GPa] are shown in Figure 8, and are valid:(2)$$\sigma =\frac{3\;\cdot \;F\;\cdot \;l}{2\;b\;\cdot h^{2}}$$
where *F* [N] is the loading force and *l* [mm] is the distance between the supports. To calculate the elastic modulus:(3)$$E=\frac{F\;\cdot \;l^{3}}{4\;\cdot \;s\;\cdot h\;\cdot b^{3}}$$
where *s* is the deflection [mm].

## 3. Results and Discussion

### 3.1. Analysis and Morphology of Fibers

**SEM images.** Together with XPS analysis, these images helped select a suitable filler type for the fabrication of the composite samples. Comparing the SEM images of RCF in Figure 1, we can see the fiber surface destruction in the plasma-treated fibers that occurred during MW plasma treatment after 10 min exposure time, similar to [12,17,18]. After MW plasma treatment after an exposure time of 1 min, deepening and highlighting the striated fiber surface occurred. Similar highlighting of the striated surface was also reported in [19,20,21].

Comparing the SEM images of RCF, Figure 5a with the MW plasma-treated images in Figure 5b and the RF plasma-treated images in Figure 5c, we can see that after MW plasma treatment, the striated surface of the fibers was highlighted. After both two types of plasma treatments, there was a loss of minor impurities formed on the recycled fibers during the recycling process. The SEM images (Figure 5, at the bottom) show the ends of the filaments, and when compared with each other no significant changes are apparent. The original assumption of the effect of plasma treatments on cross sections and fiber ends was not observed. However, the removal of impurities is noticeable.

**TEM images.** Due to the nature of TEM imaging, we focused on the ends of the fibers, where details of the longitudinal arrangement of the carbon fiber surface can be seen. No significant changes were observed between fibers with and without plasma treatment at the fiber ends. In Figure 6, the turbostratic structure of the transverse fracture arrangement of the fibers can be seen. Therefore, the original assumption of the influence of plasma treatments on the cross sections and fiber ends was not confirmed.

**Measurement of wettability.** The contact angle of WCA recycled carbon fiber is affected by plasma action. Figure 7 shows that it can be observed that the plasma treatment has a significant effect on the changes in the hydrophobic character of the recycled carbon fibers; both the type of plasma treatment and the duration of this treatment influence results. In the case of MW plasma, the WCA value decreased with a longer plasma treatment time. Untreated fibers exhibit hydrophobic behavior, while plasma treated fibers exhibit hydrophilic behavior.

Due to the roughness of the surface, the measured contact angle is only indicative of the plasma treatment trend. The measured values show that plasma treatment of CF has a significant effect on the change of hydrophilic properties. Untreated fibers show hydrophobic behavior, while plasma treated fibers show hydrophilic behavior. The increased hydrophilicity of the fibers is probably due to the binding of oxygen functional groups to the fiber surface, together with the higher surface roughness caused by the plasma treatment.

However, in general, roughness is considered a relatively unimportant factor causing contact angle hysteresis [2]. The hysteresis is dominated by the heterogeneity of the surface chemistry; thus the reduced hysteresis may be due to the more uniform chemical composition of the plasma-treated CF surface.

Comparing the WCA results shown in Figure 7, we see that microwave plasma’s longer plasma exposure time reduces the WCA angle value, i.e., the fibers are more wettable after more extended MW treatment. Nevertheless, considering the destructions demonstrated by SEM analysis, fibers with an MW treatment time of 1 min were chosen for application to composites.

**Raman spectroscopy** was used to assess the effect of plasma treatment on the structural changes of carbon fibers. The Raman spectra (RS) for different types of carbon structures show two peaks at approximately 1355 and 1580 cm^−1^, corresponding to the defective carbon (D band) and graphite mode (G band) [22]. Figure 8 shows the RS of RCF and the plasma-treated RCF. The intensities of the D and G oscillations are related to the graphite crystal size and the proportion of the amorphous carbon phase. These peaks correspond to the usual D and G peaks in PAN-based carbon fibers reported by Fitzer and Rozploch [23]. Recycling of carbon fibers does not change their RS spectra and thus probably does not significantly change their surface structure. Comparing the RS results of RCF and those of plasma-treated RCF, it can be concluded that plasma treatment did not fundamentally change the crystalline structure of RCF because plasma only produces a surface effect in the nanometer range.

**XPS analysis.** The plasma treatments modify surface of carbon fibers significantly. The main observed effect is functionalization of fibers by introducing oxygen functionalities on the surface. The plasma treatment increased oxygen content twice from 10% on untreated fibers up to approximately 20% on plasma treated in both types of discharges. The nitrogen content is not significantly changed by the plasma treatment.

The detailed study of chemical bonding on the surface was made on high resolution photoelectron spectra, see Figure 9. Carbon related photoelectron peak C1s consists of multiple components because carbon bonds to carbon, nitrogen and oxygen are expected. There are many possible types of bonds, thus the components shall be interpreted as not comprehensive list. The main components were identified as follow: C–C at 284.5 eV, C–O–C ether-like at 286.2 eV, C=O carbonyl-like at 287.4 eV, C=O–O carboxyl/ester-like at 288.1 eV and p-p shake-up at 291.8 eV in agreement with literature [10,24]. The C to N bonds should also be present in C1s peak but due to the wide range of possible binding energies of this components, and lower content of nitrogen compare to oxygen, we have neglected C–N bonds in C1s spectra; moreover, the C–N components are overlapped by C–O components.

Literature reported Raman analyses showed both G and D bands typical for sp2 and defects/sp3 hybridization [25]. Therefore, we have introduced a sp3 related component to the C1s component model at binding energy about 0.8 eV shifted from the main C–C peak, then the original C–C peak acts as a sp2 related component. When the model without the sp3 component was used, the C–C component was approximately in the amount of sp2 + sp3 components. The Raman analyses in literature [25] shows an increase in ratio D/G for the plasma treated samples and a decrease for thermal treated samples. Although the C1s component for sp3 and sp2 hybridization are not equal to the D and G bands, we observe similar behavior for the MW plasma treatment ratio sp3/sp2 from XPS spectra is increasing from 0.23 up to 0.7 when RF plasma treatment leads to a decrease in ratio sp3/sp2 to 0.17 for some conditions. The RF plasma treatment is similar to thermal treatment for some materials [26]. This is probably because higher energy particles are created in the discharge and they can modify the fibers more inside with more effective heating effect than low-energy particles in the MW plasma, with dominant surface effect only.

The nitrogen related N 1 s peek has two components [24] located at around 398.5 eV and 400.5 eV binding energy. Nitrogen present in the carbon fibers is from the polyacrylonitrile (PAN) used as precursor. Even after carbonization at temperatures above 1000 °C a small amount of nitrogen remains in the fibers, and it is mostly incorporated in the graphite-like structure in two ways: either at the center of three aromatic rings, bonded to three carbon atoms (graphitic nitrogen); or bonded to two carbon atoms at the edges of the graphite-like sheets or at defects (pyridine-like nitrogen). XPS study shows significant differences between plasma treated fibers [24]. The lower binding energy component represents the pyridine-like nitrogen and this type of nitrogen is reduced by both of the studied plasma treatments. The pyridine-like nitrogen is reduced from an initial 23% to 19% in 100 W and longer 30 W RCF plasma-treated RF, and to 12% in RCF plasma-treated MW for treatment times longer than 30 min. Therefore, we can control the N1s content on the surface in this range by plasma conditions.

The oxygen photoelectron peek O1s can be successfully fitted by three components. Where the component at lowest energy 531 eV is O=C, component at 532.9 eV is O–C similarly as in polymers [24], the final weak component about 536.4 eV is adsorbed water. The plasma treatments are significantly reducing the amount of lowest energy component O=C from an initial 40% to approximately 17% for MW or 20% for RF plasma. The simultaneously O–C component is increased in relative and also in absolute values because of the increase in the total amount of oxygen.

### 3.2. Characteristics of FRE Composites

**Characteristics of fracture surfaces.** SEM analyzed the surface of the fracture surfaces after impact toughness measurements. Figure 10a is a snapshot of the pure resin without filler. Figure 10b–d are microphotographs of the composites filled with recycled carbon fibers at 1 wt%.

After impact toughness measurements on FRE composite specimens, the fracture surface images obtained by SEM were analyzed. The dispersion state of the recycled carbon fibers in the epoxy resin was evaluated. Figure 10a, pure resin, shows a brittle homogeneous fracture surface. The images of the composites containing filler in the form of RCF, in Figure 10b–d, show fibers that are dispersed continuously in the matrix. According to morphology of the carbon fibers, these short fibers can transmit stress, which leads to the absorption of external energy during mechanical testing. When cracks form in the composite, the dispersed fibers of the modified RCFs can prevent the propagation of these cracks, but only to a certain extent and at lower fiber concentrations, as also reported by Altay [10]. The improved mechanical properties of epoxy composites with plasma modified RCFs are due to the inhibition of crack growth, which was also confirmed in [27,28].

**Impact toughness of FRE composites.** The impact toughness results are shown in Figure 11. One of the possible disadvantages of using micro fillers in epoxy matrices is the potential formation of microscopic cracks. Microcracks in the materials can propagate under repeated loading or a combination of different loads and lead to the destruction of the composite, as pointed out by Teh [29]. If the measured values of the impact toughness of composites filled with recycled carbon fiber are compared with the impact toughness of pure resin (in Figure 11), the RCF plasma treatment improves the impact toughness of the samples with 1 wt% RCF. In contrast, plasma treatment of fibers has no significant effect on samples with high filler concentrations. Low-fiber concentrations (1 wt%–2.5 wt%) increased the tested composites’ impact toughness values compared to the unfilled resin’s impact toughness values. Compared to other studies [27,28,29], there was no stable increase in impact toughness values for the composites filled with recycled carbon fibers; the composites with the lowest filler concentration achieved the best results, while higher concentrations resulted in a deterioration of impact toughness compared to the impact toughness value of unfilled resin. This is probably due to the shape of the filler in the form of cylinders, and it may also be due to the formation of clusters after reaching a suitable fiber concentration. In contrast, at higher fiber concentrations, the influence of micro-layers reduces the impact toughness value, as also pointed out by [28].

**Three-point bending. The three-point bending results are shown in Figure 12.** Flexural strength is the ability of material to resist a bending force applied perpendicular to the longitudinal axis of the composite specimen. The stress induced by the bending load is then a combination of compressive and tensile stresses [29]. Figure 12 compares the tested composite specimen elastic modulus and bending stress values. It can be seen that the addition of fibers to the resin improves the elastic modulus and bending stress for all fiber concentrations and types. The effect of plasma treatments is negligible except at the lowest concentrations. At low concentrations of 1 wt%, an increase in elastic modulus is observed, namely by 10% for samples with RCF fibers without plasma treatment, by 29% for samples with RCF with RF plasma treatment, and 21% for samples with RCF with MW plasma treatment. The improvement in bending stress values at the same concentration was 15% for samples with RCF without plasma treatment, 31% for samples with RCF with RF plasma treatment, and 26% for samples with RCF with MW plasma treatment. The scatter of the elastic modulus and bending stress values is lower for samples with fibers treated with both types of plasma treatments than for samples with untreated fibers. In this study, the samples are prepared using magnetic and manual mixing, which can be a problem at higher-fiber concentrations and can lead to increased fiber agglomeration and, indirectly, higher values of flexural modulus.

## 4. Conclusions

Fiber-reinforced polymers, particularly carbon fiber reinforced epoxies, are expected to account for more than 50% of the structural weight of motor vehicles in the future. As the number of applications increase, the amount of ancient carbon fiber structures will grow, and recycling of these materials will become more important [2]. Therefore, in this work, attention was paid to plasma modifications of short RCFs, to determine the suitability of applying these modifications to improve the mechanical properties of epoxy composites filled with these modified fibers.

This study analyzed the mechanical properties (represented by impact toughness and three-point bending) of epoxy composites filled with short RCFs. Two types of plasma treatment were used to modify the RCF surface. These treatments were performed using radio frequency (RF) and microwave (MW) radiation at the same pressure and atmosphere. WCA measurements and XPS analysis confirmed that plasma treatment changes the surface layers of carbon fibers, as reported in studies [10,11]. The SEM and TEM images reported in this study focused on the region of the fiber ends, and their cross-sections, since these areas form a significant part of the interfacial interface in short fiber-filled composites. As expected, based on the pilot study [12], the use of both types of plasma treatment increases the wettability of the recycled carbon fibers. Both types of plasma treatments also contribute to the ease of fiber introduction into the epoxy matrix.

The suitability of plasma treatment was verified in mechanical tests performed on composite samples prepared from these fibers. It was found that the modulus and bending stress increased with increasing recycled carbon fiber content, as also reported in the study [10,11]. However, the difference in the effect of plasma treatment on the increase in modulus and bending stress for samples with higher concentrations are negligible, compared to the data reported in [10,11]. This is probably due to the effect of the technology used to prepare the composite samples, or in this study, the samples are prepared using magnetic and manual mixing, which can be a problem for higher fiber concentrations and can lead to increased fiber agglomeration; whereas in [10,11] injection molding technology is used, but this is also due to the type of polymer matrix (PP) used.

Furthermore, in this work, the spectrum of different concentrations of epoxy-filled RCF composites has been analyzed. The increase in the adhesion between fibers and matrix is due to the increase in the surface roughness of the fibers and the oxidation of their surface. For impact toughness, it was found that there is an optimum limit. A suitable balance between elasticity and toughness was achieved for composite samples with 1 wt% RCF. and both types of plasma treatment further improved the tested mechanical properties by this concentration of RCF. However, RF plasma treatment of the fibers had a more significant effect on the improvement of impact toughness and modulus.

The selected RF plasma treatment procedure is readily applicable for plasma treatment of short recycled fibers and represents a significant advantage (for the lowest concentration samples) over the use of untreated RCF. FRE composites reinforced with plasma-modified RCF fibers can be obtained for various applications in the composites industry.

## Figures and Tables

**Figure 1 materials-15-06290-f001:**
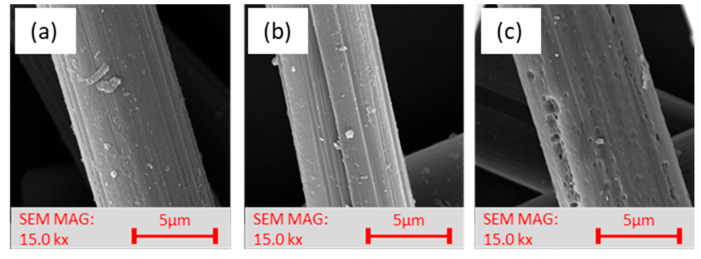
SEM photos of RCF: (**a**) without treatment; (**b**) MW plasma-treated for 1 min; and (**c**) MW plasma-treated for 10 min.

**Figure 2 materials-15-06290-f002:**
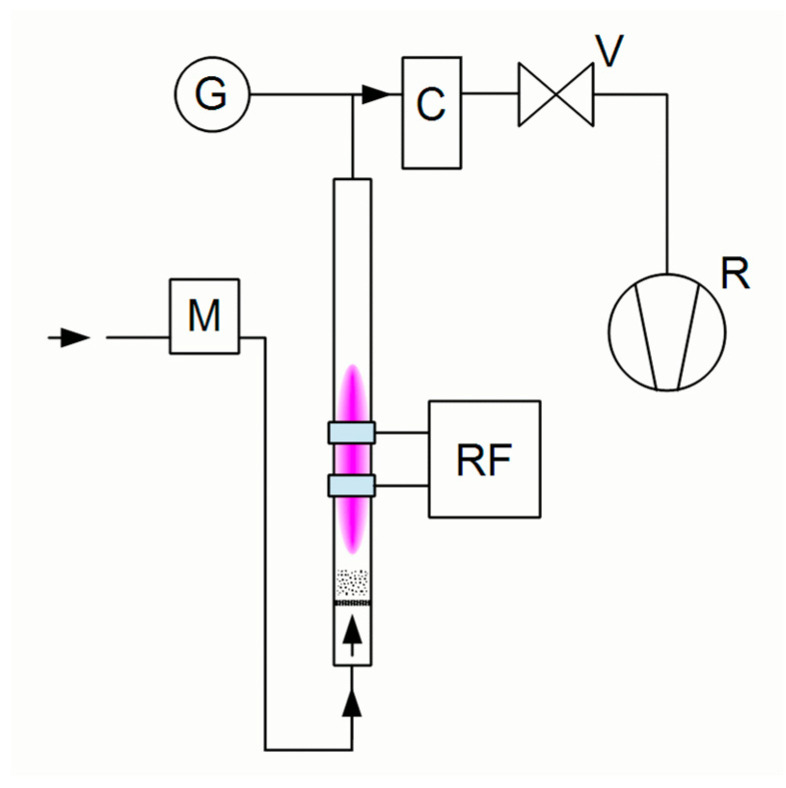
Diagram of RF reactor: G—vacuum gauge, C—cyclone, V—valve, M—mass flow controller, RF—generator with matching unit, R**–**rotary vane pump.

**Figure 3 materials-15-06290-f003:**
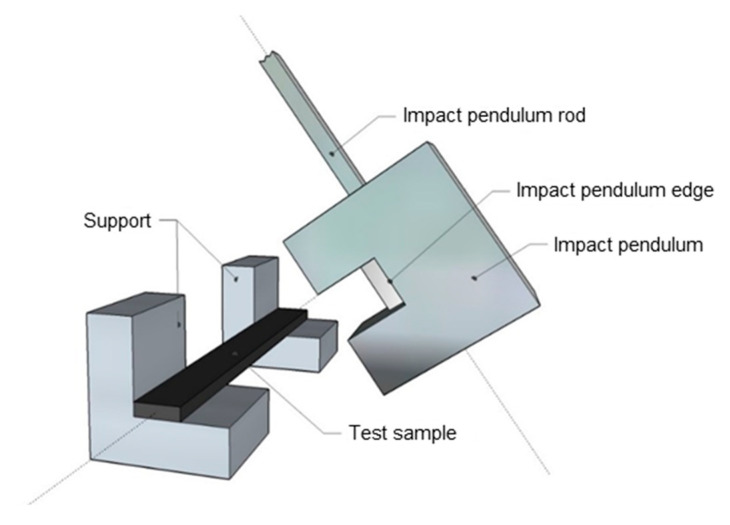
Schematic of the Charpy impact test.

**Figure 4 materials-15-06290-f004:**
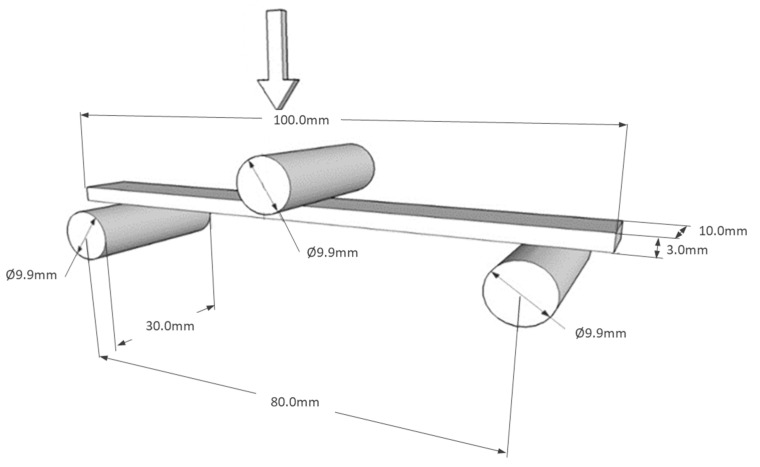
Schematic of the three-point bending test.

**Figure 5 materials-15-06290-f005:**
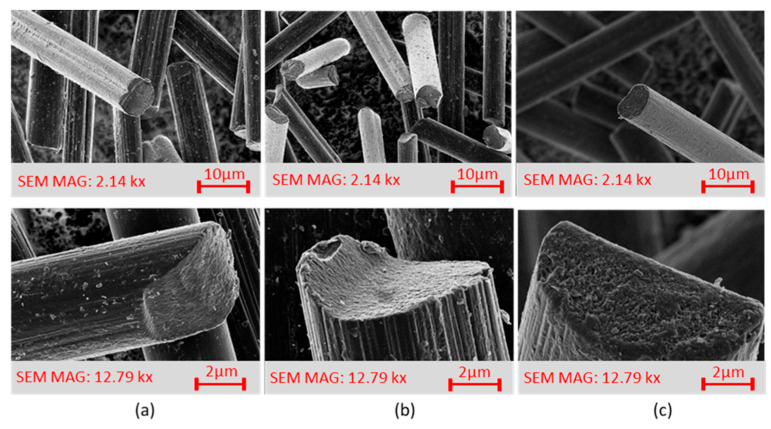
SEM photos of RCF (longitudinal views in the upper part of the figure, cross-sectional details in the lower part of the figure): (**a**) RCF; (**b**) MW plasma-treated for 1 min; and (**c**) RF plasma-treated for 30 s.

**Figure 6 materials-15-06290-f006:**
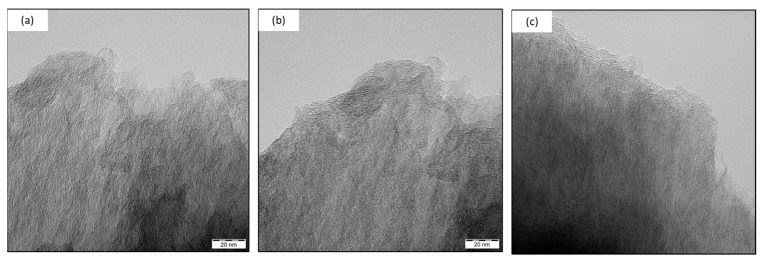
TEM images of the ends of the used RCF: (**a**) RCF; (**b**) MW plasma-treated for 1 min; and (**c**) RF plasma-treated for 30 s.

**Figure 7 materials-15-06290-f007:**
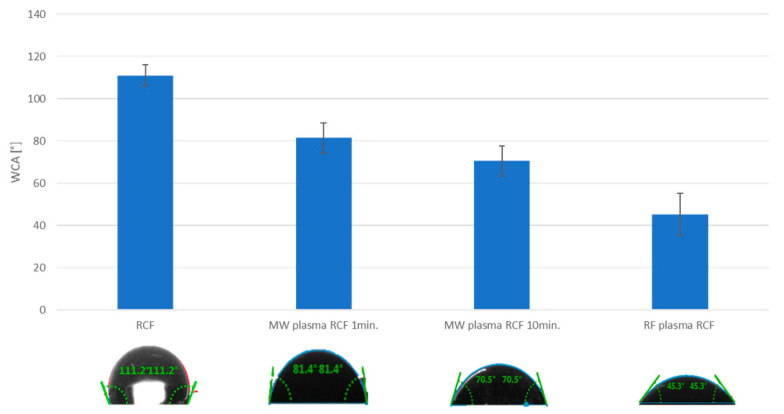
Measurement of wettability of RCF.

**Figure 8 materials-15-06290-f008:**
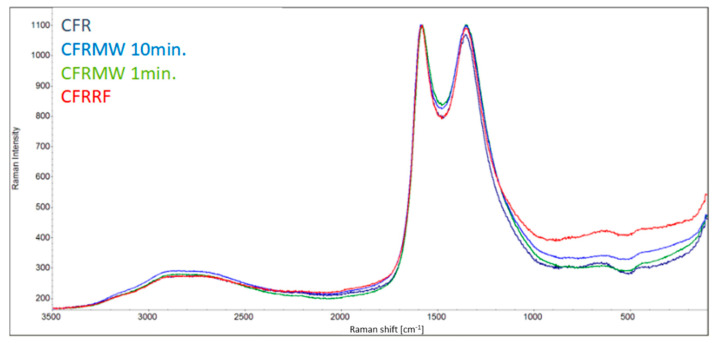
RS used RCF.

**Figure 9 materials-15-06290-f009:**
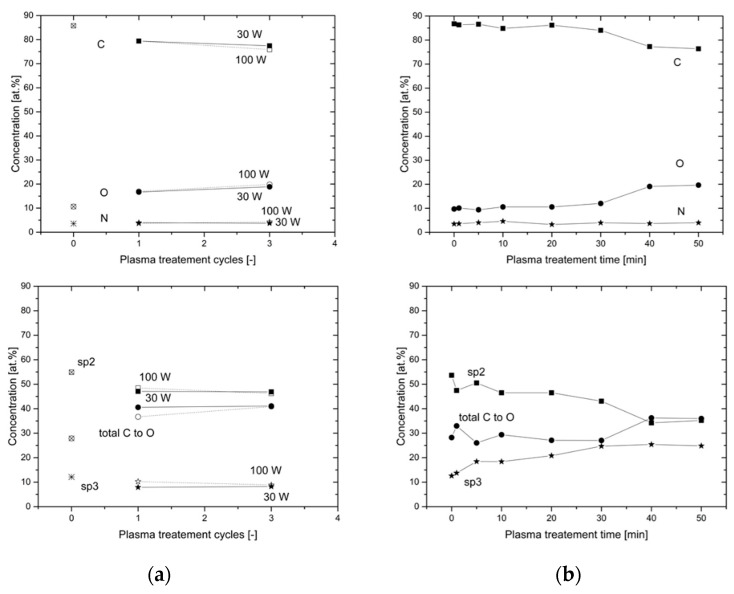
XPS analysis of plasma-treated RCF: (**a**) RF plasma-treated (dotted line = fibers treated at 30 W, solid line = fibers treated at 100 W); and (**b**) MW plasma-treated.

**Figure 10 materials-15-06290-f010:**
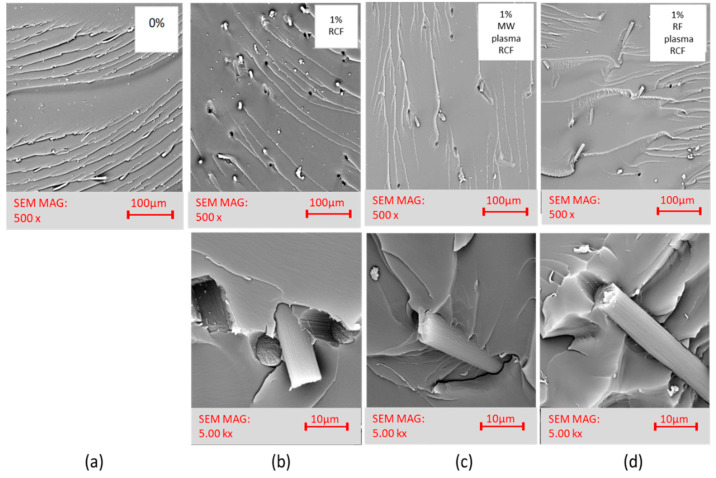
SEM photos of FRE after impact toughness measurements (view of the total fracture surface in the upper part of the figure, details of the fiber-resin interfacial interface in the lower part of the figure): (**a**) neat resin; (**b**) RCF–1 wt%; (**c**) MW plasma-treated RCF–1 wt%; and (**d**) RF plasma-treated RCF–1 wt%.

**Figure 11 materials-15-06290-f011:**
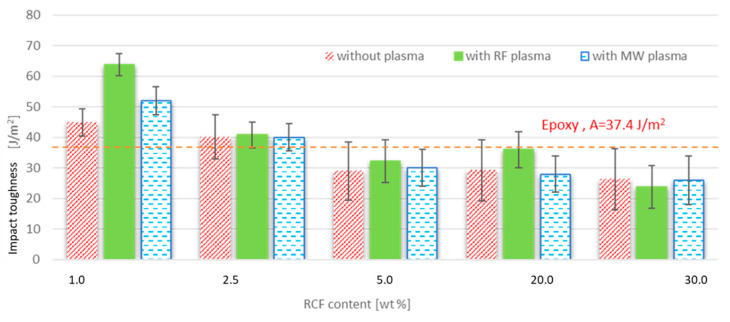
The impact toughness of FRE composites.

**Figure 12 materials-15-06290-f012:**
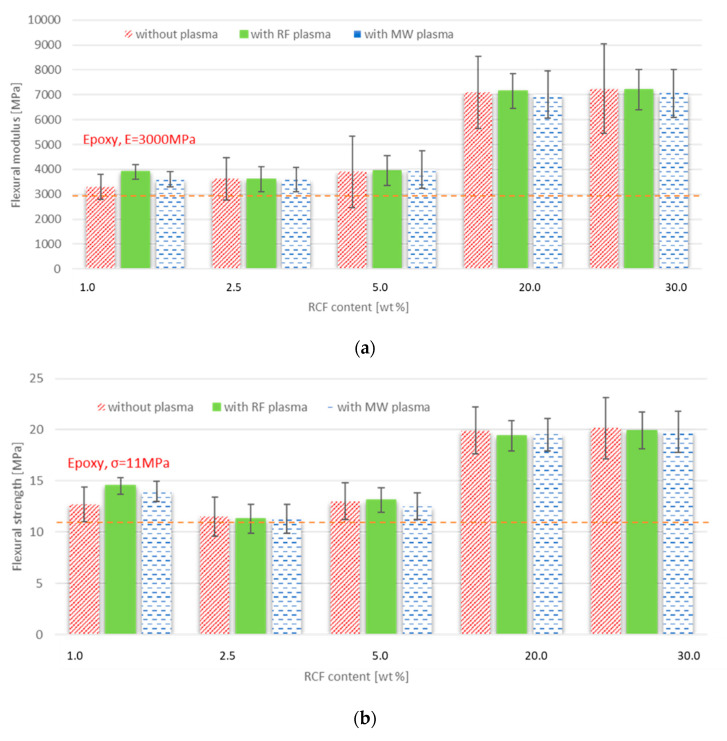
The flexural strength of FRE composites: (**a**) flexural modulus; and (**b**) flexural strength.

## Data Availability

Not applicable.

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
