# Peer review of "Comparison of the Influence of Two Types of Plasma Treatment of Short Carbon Fibers on Mechanical Properties of Epoxy Composites Filled with These Treated Fibers"

_materials, 2022, doi:10.3390/ma15186290_

Round 1
Reviewer 1 Report
The authors investigated the influence of two types of plasma treatment (radio and microwave frequencies) of short carbon fibers with gradient concentrations on mechanical properties of epoxy composites filled with these treated fibers. There are some suggestions to note:
1. Abbreviations of proper nouns appearing in the abstract should be written in full at the first appearance, such as XPS and SEM, etc.
2. Are there any studies reporting the effect of plasma treatment on the mechanical properties of carbon fiber epoxy composites? Please briefly describe in the introduction.
3. Why choose these two mechanical parameters (impact toughness and flexural strength)?
4. Please supplement the test setup diagrams and specimen details for impact toughness and flexural strength tests.
Author Response
We have tried to complete all the necessary data and to incorporate the comments of all three reviewers into the edits. The figure numbering has been changed, three figure diagrams have been added, two sources have been added to the references, and the introduction and conclusion of the paper have been completely changed and revised. Thank you very much for taking the time to read the paper.

Reviewer 2 Report
This study investigated the effects of two plasma treatments on short carbon fibers. A systematic experimental study on plasma treatment of short carbon fiber was carried out. The fibers were tested by scanning electron microscopy, Raman microscopy, transmission electron microscopy, X-ray photoelectron spectra and wetting measurements. The stiffness, strength and impact toughness of the fiber reinforced composites were tested. Different fiber volume content and plasma treatment methods were studied. The experiment of this study is complete, but lacks innovation. The paper needs very significant improvement before acceptance for publication, especially the significance of the research in the introduction. My detailed comments are as follows:
(1) Page 1, line 17. What is XPS analysis? The full name of XPS should be written when it first appears.
(2) The introduction does not reflect the innovation of the study. What is the difference between this study and previous plasma treatment studies for virgin CF [5-8] or recycled RCFs [9-11]? What are the shortcomings of the previous studies? What is the significance of this study?
(3) Page 2, line 59. I guess the short carbon fibers of this study is obtained by grinding the virgin carbon fibers. Short carbon fibers should not be called recycled carbon fibers. Recycled carbon fibers are made from carbon fiber reinforced plastics. According to the research of Borjan et al., there are three methods to obtain recycled carbon fibers. Therefore, your research object is short carbon fiber, not recycled carbon fiber. It is recommended to replace the recycled carbon fiber with short carbon fiber. “Currently, chemical, mechanical, and thermal methods have been used to recycle composites. A chemical process is used to recover both the clean fibers and fillers and depolymerized matrix in monomers or petrochemical feedstock. Mechanical recycling is mainly based on crushing, grinding, milling, and shredding the composite part into smaller pieces, which can then be further ground into powder. Thermal recycling involves heat breaking the scrap composite and combusting the resin matrix, thereby recovering the carbon fibers.” —Borjan, D., Knez, Ž., & Knez, M. (2021). Recycling of carbon fiber-reinforced composites—difficulties and future perspectives. Materials, 14(15), 4191. https://doi.org/10.3390/ma14154191
(4) Page 2, line 71 and page 3, line 89. Could you provide a schematic diagram of the fluidization device and fluidized bed reactor? Is the short carbon fiber plasma treated on a flat plate or in a roller?
(5) Page 5, line 171. “In Fig. 3, the turbostratic structure of the transverse fracture arrangement of the fibers can be seen, and it can be assumed that the original fibers had a moderately high modulus of elasticity.” How to get the conclusion of moderately high modulus of elasticity. There is no direct relationship between the turbostratic structure and high modulus.
(6) Page 5, line 177. Is there any relation between the hydrophobic character and resin wettability? Can water represent the resin matrix?
(7) Page 6, line 198. “Recycling of carbon fibers does not change their RS spectra”. The word "recycling" and "recycle" are not found in reference [20]. This conclusion is misquoted.
(8) Page 9, line 267. “After mechanical tests on FRE composite specimens, the fracture surface images obtained by SEM were analyzed.” The mechanical test refers to three-point bending test or impact toughness test?
(9) Page 1 Abstract. “It was found that the modulus of elasticity and bending stress increase with the increasing content of recycled carbon fibers.” This conclusion is well known. Please replace it with a more valuable conclusion.
(10) Page 10, line 322, conclusion. “plasma treatments change the surface layers” It's too vague. You should briefly explain what has changed on the surface layers of carbon fibers.
(11) Too many abbreviations, please reduce the use of abbreviations.
(12) The conclusion does not reflect important work contents. The important phenomena observed in each test should be briefly explained in the conclusion.
(13) Is page 7 an Analysis of Figure 6? If yes, please explain as shown in Figure 6.
(14) Page 10 line 314. The variance of the elastic modulus is generally small, but figure 9 shows that the variance of the elastic modulus reaches about 20%, which is different from my experience. How the displacement is measured in the three-point bending test.
(15) Page 4, line 139. What is the thickness of the test specimen?
(16) Page 4 line 144 and 147. The meaning of h and b in equation 2 and 3 is wrong. They should be interchanged. Introduction to three-point bending in Wikipedia: https://en.wikipedia.org/wiki/Three-point_flexural_test
Author Response

(The authors gave the same response as above.)

Reviewer 3 Report
1. It is unclear whether ‘PAN based Carbiso milled carbon fibre’ is a recycled one. If yes, I recommend using ‘PAN based Carbiso milled recycled-carbon fibre.’ (line 58)
Moreover, ‘PAN based’ is needed to change to ‘PAN-based’ and ‘fibre’ is also to ‘fiber.’
2. (Line 177) The surface characteristics of carbon fibers were dramatically changed to hydrophilic after plasma treatments. Please add more discussion about the mechanism.
3. (Line 191) There are significant changes in D and G bands (even in the amorphous area) according to the plasma conditions. Please add more discussion about it.
4. I cannot find the C1s, N1s, and O1s graphs though the discussion is in the manuscript.
5. Naming style of all samples in the manuscript must be unified.
6. Please add (a) and (b) in Figure 9 and use the full name on the Y-axis of both graphs.
7. In the three-point bending test, there is no significant change before and after the plasma treatment except for the 1% carbon fiber sample. Moreover, the authors did not describe the exact reasons for this behavior.
Author Response

(The authors gave the same response as above.)

Round 2
Reviewer 3 Report
This paper has been well-revised according to my comments.